# LEARNING NUMERAL EMBEDDING

## ABSTRACT

Word embedding is an essential building block for deep learning methods for natural language processing. Although word embedding has been extensively studied over the years, the problem of how to effectively embed numerals, a special subset of words, is still underexplored. Existing word embedding methods do not learn numeral embeddings well because there are an infinite number of numerals and their individual appearances in training corpora are highly scarce. In this paper, we propose two novel numeral embedding methods that can handle the out-of-vocabulary (OOV) problem for numerals. We first induce a finite set of prototype numerals using either a self-organizing map or a Gaussian mixture model. We then represent the embedding of a numeral as a weighted average of the prototype number embeddings. Numeral embeddings represented in this manner can be plugged into existing word embedding learning approaches such as skip-gram for training. We evaluated our methods and showed its effectiveness on four intrinsic and extrinsic tasks: word similarity, embedding numeracy, numeral prediction, and sequence labeling.

## 1 INTRODUCTION

Word embeddings, the distributed vector representations of words, have become the essential building block for deep learning approaches to natural language processing (NLP). The quality of pretrained word embeddings has been shown to significantly impact the performance of neural approaches to a variety of NLP tasks. Over the past two decades, significant progress has been made in the development of word embedding techniques (Lund & Burgess, 1996; Bengio et al., 2003; Bullinaria & Levy, 2007; Mikolov et al., 2013b; Pennington et al., 2014). However, existing word embedding methods do not handle numerals adequately and cannot directly encode the numeracy and magnitude of a numeral (Naik et al., 2019). Most methods have a limited vocabulary size and therefore can only represent a small subset of the infinite number of numerals. Furthermore, most numerals have very scarce appearances in training corpora and therefore are more likely to be out-of-vocabulary (OOV) compared to non-numerical words. For example, numerals account for $6.15\%$ of all unique tokens in English Wikipedia, but in GloVe Pennington et al. (2014) which is partially trained on Wikipedia, only $3.79\%$ of its vocabulary is numerals. Previous work (Spithourakis et al., 2016) also shows that the numeral OOV problem is even more severe when learning word embeddings from corpora with abundant numerals such as clinical reports. Even if a numeral is included in the vocabulary, its scarcity in the training corpus would negatively impact the learning accuracy of its embedding.

The inadequate handling of numerals in existing word embedding methods can be problematic in scenarios where numerals convey critical information. Take the following sentences for example,

"Jeff is ***190***, so he should wear size XXL" (190 is a reasonable height for size XXL)

"Jeff is ***160***, so he should wear size XXL" (160 is an unreasonable height for size XXL)

"Jeff is ***10***, so he should wear size XS" (10 is an age instead of a height)

If the numerals in the example are OOV or their embeddings are not accurately learned, then it becomes impossible to judge the categories of the numerals or the reasonableness of the sentences.

In this paper, we propose two novel methods that can produce reasonable embeddings for any numerals. The key idea is to represent the embedding of a numeral as a weighted average of a small set of prototype number embeddings. The prototype numerals are induced from the training corpus

using either a self-organizing map (Kohonen, 1990) or a Gaussian mixture model. The weights are computed based on the differences between the target numeral and the prototype numerals, reflecting the inductive bias that numerals with similar quantities are likely to convey similar semantic information and thus should have similar embeddings. Numeral embeddings represented in this manner can then be plugged into a traditional word embedding method for training. We empirically evaluate our methods on four tasks: word similarity, embedding numeracy, numeral prediction, and sequence labeling. The results show that our methods can produce high-quality embeddings for both numerals and non-numerical words and improve the performance of downstream tasks.

## 2 RELATED WORK

**Word Embedding** Word embeddings are vector representations of words which carry semantic meanings implicitly and are trained without supervision. Most existing word embedding training methods can be divided into two classes. The first class of methods (Lund & Burgess, 1996; Rohde et al., 2006; Bullinaria & Levy, 2007; Lebret & Lebret, 2013) extract word co-occurrence statistics from the training corpus, compute a word-word matrix based on measures such as PPMI, and then apply dimension reduction techniques such as principle component analysis to produce a low-dimensional vector representation for each word. The second class of methods (Bengio et al., 2003; Collobert & Weston, 2008; Mikolov et al., 2013a;b) use a simple neural network to model the relation between a word and its context within a sliding window in the training corpus. GloVe (Pennington et al., 2014) has been proposed as a method that combines the advantages of both classes. All the above methods have a finite vocabulary size and use a 'UNK' symbol to represent OOV words. Recent work (Naik et al., 2019) shows that these popular methods do not handle numerals adequately. Wallace et al. (2019) shows that existing word embedding methods can encode numeracy implicitly for high-frequency numerals, but the embedding's numeracy for OOV numerals is not investigated. Our goal is to design better numeral embedding methods that can be integrated into traditional word embedding methods and handle the OOV problem for numerals.

**Numeracy in natural language** Numeral understanding has been found important in textual entailment (Lev et al., 2004; De Marneffe et al., 2008; Roy et al., 2015) and information extraction (Intxaurrondo et al., 2015; Madaan et al., 2016), but existing systems often use manually defined task-specific features and logic rules to identify numerals, which is hard to generalize to other tasks. A lot of research has been done trying to solve math problems, using either manually designed features and rules (Roy et al., 2015; Mitra & Baral, 2016; Roy & Roth, 2016; Upadhyay et al., 2016) or sequence-to-sequence neural networks Wang et al. (2017), but the quantity of numerals is not important in this task and hence existing methods often replace numerals by dummy symbols such as $n_1$ and $n_2$. Spithourakis & Riedel (2018) studied different strategies to better model numerals in language models. Chen et al. (2019) created Numeracy-600K dataset and studied the ability of neural network models to learn numeracy. Our work differs from previous work in that we aim to produce general-purpose numeral embeddings that can be employed in any neural NLP approach.

## 3 METHODS

Given a training corpus $C$, we first extract all the numerals using regular expressions and form a dataset $X$ containing all the numbers represented by these numerals. A number (e.g., 2000) may appear for multiple times in $X$ if its corresponding numerals (e.g., '2000', '2,000', etc.) appear for multiple times in $C$. We then induce a finite set $\mathbb{P}$ of typical numerals (i.e., prototypes) from $X$ using a self-organizing map (Kohonen, 1990) or a Gaussian mixture model. We also define a function $sim(n_1, n_2)$ outputting the similarity between two arbitrary numbers $n_1$ and $n_2$. Now we represent the embedding of any target numeral $n$ as a weighted average of the prototype number embeddings with the weights computed by the similarity function:

$$e(n) = \cdot \sum_{p \in \mathbb{P}} \alpha \cdot sim(n, p) \cdot e(p), \quad \sum_{p \in \mathbb{P}} \alpha \cdot sim(n, p) = 1 \qquad (1)$$

We use $e(\cdot)$ to denote the embedding of a number $\alpha$ is the normalization factor. This formulation satisfies the intuition that numerals with similar quantities are likely to convey similar semantic information and thus should have similar embeddings.

Our numeral embeddings can be integrated into traditional word embedding methods such as skip-gram for training. During training, we back-propagate the error gradient to update the prototype number embeddings. In this way, the prototype number embeddings (and hence all the numeral embeddings) are learned jointly with non-numerical word embeddings.

## 3.1 Squashing Numbers to Log-space

Inspired by psychological evidence that our brain compresses large quantities nonlinearly using a logarithmic scale on the mental number line (Nieder & Miller, 2003; Dehaene, 2011), we design the following squashing function to transform all the numbers in $X$ into the log-space before prototype induction. Alternatively, we can apply the function only in the similarity function. Besides the psychological motivation, squashing is also necessary for our methods to avoid overflow during training when there are very large numbers such as $10^{15}$ in the training corpus.

$$f(x) = \begin{cases} \log(x) + 1, & \text{if } x > 1 \\ x, & \text{if } x \in [-1, 1] \\ -\log(-x) - 1, & \text{if } x < -1 \end{cases} \quad (2)$$

## 3.2 Prototype Induction

We develop two methods for inducing a small prototype set $\mathbb{P}$ from the number dataset $X$. Denote the number of prototypes by $m$.

**Self-Organizing Map** A self-organizing map (SOM) (Kohonen, 1990) is an artificial neural network that can be viewed as a clustering method. After training a SOM on the dataset $X$, we regard each cluster centroid as a prototype. One advantage of using a SOM in comparison with traditional clustering methods is that it distributes prototypes more evenly on the number line and may assign prototypes to number ranges with few training samples, which we expect would lead to better generalizability.

**Gaussian Mixture Model** Inspired by psychological study of the mental number line (Dehaene et al., 2003) and previous work on language modeling (Spithourakis & Riedel, 2018), we train a Gaussian mixture model (GMM) to induce number prototypes. A GMM is defined as follows.

$$p(U = n) = \sum_{k=1}^{m} P(Z = k)P(U = n|Z = k) = \sum_{k=1}^{m} \pi_k \mathcal{N}(n; \mu_k, \sigma_k^2) \quad (3)$$

where $Z$ is a latent variable representing the mixture component for random variable $U$, and $\mathcal{N}$ is the probability density function of a normal distribution, and $\pi_k, \mu_k, \sigma_k \in \mathbb{R}$ represent the mixing coefficient, mean and standard deviation of the $k$-th Gaussian component. We train a GMM on the number dataset $X$ using the expectation-maximization (EM) or hard-EM algorithm and regard the means of the learned Gaussian components as our prototypes $\mathbb{P} = \{\mu_1, \cdots, \mu_m\}$. We use three GMM initialization methods described in Appendix A.

## 3.3 Similarity Function

For SOM-induced prototypes, we define the following similarity function:

$$sim(p, n) = |g(p) - g(n)|^{-\beta}, \quad \beta > 0, \ p \in \mathbb{P} \quad (4)$$

where function $g$ is equal to the squashing function $f$ defined in Eq.2 if we do not apply log transformation before prototype induction and is the identity function $I$ otherwise. $\beta$ is a hyper-parameter set to $1.0$ by default.

For GMM-induced prototypes, we can naturally use the posterior probability of the component assignment to define the similarity function.

$$sim(p_k, n) \propto P(Z = k|U = n) = \frac{\pi_k \mathcal{N}(n; \mu_k, \sigma_k^2)}{\sum_{k=1}^{m} \pi_k \mathcal{N}(n; \mu_k, \sigma_k^2)} \ , \quad p_k \in \mathbb{P} \quad (5)$$

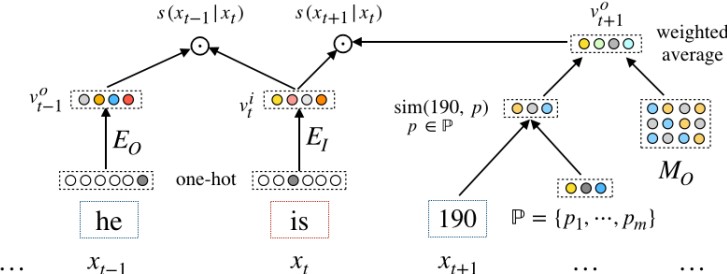

Figure 1: The computational graph when the center word is 'is' and the context words are 'he' and the numeral '190'. We look up the embedding vectors of non-numerical words directly from the embedding matrices and use the weighted average of prototype embeddings as the numeral embedding. Negative sampling is not shown in the figure.

### 3.4 EMBEDDING TRAINING

We now describe how to integrate our numeral embeddings into traditional word embedding methods for training. We choose skip-gram with negative sampling (Mikolov et al., 2013a;b) as the word embedding method here, but many other word embedding methods such as CBOW (Mikolov et al., 2013a), HAL (Lund & Burgess, 1996) and GloVe (Pennington et al., 2014) can be used as well.

Skip-gram is a word embedding method based on the idea of context word prediction. The training corpus $C$ is regarded as a sequence of words $(x_1, \ldots, x_T)$. For token $x_t$, we define the preceding and following $c$ tokens as the context of $x_t$. Skip-gram aims to maximize $p(x_{t+j}|x_t)$ $(-c \leq j \leq c)$, the probability of a context word given the center word $x_t$. To formulate $p(x_{t+j}|x_t)$, skip-gram associates each word $x_i$ with two vector representations: the input embedding $v_{x_t}^i$ for being a center word and the output embedding $v_{x_t}^o$ for being a context word. The input and output embeddings of all the words in the vocabulary $\mathbb{V}$ constitute matrices $E_I \in \mathbb{R}^{D \times |\mathbb{V}|}$ and $E_O \in \mathbb{R}^{D \times |\mathbb{V}|}$ respectively, where $D$ is the dimension of word embeddings. The conditional probability $p(x_{t+j}|x_t)$ is then defined to based on the dot product $s(x_{t+j}|x_t) = v_{x_t}^i{}^T v_{x_{t+j}}^o$. Nagative sampling is used to approximate the normalization factor for the conditional probability.

$$\log p(x_{t+j}|x_t) \approx \log \sigma(v_{x_{t+j}}^o{}^T v_{x_t}^i) + \sum_{i=1}^{k} \mathbb{E}_{x_i \sim P_n(x)} [\log \sigma(-v_{x_i}^o{}^T v_{x_t}^i)] \qquad (6)$$

where $\sigma$ denotes the sigmoid function, and $P_n(x)$ is the negative word sampling distribution used to draw $k$ negative samples.

We modify skip-gram by computing numeral embeddings differently from non-numerical word embeddings. We associate each prototype number with an input embedding and an output embedding. The input and output embeddings of all the prototypes constitute matrices $M_I \in \mathbb{R}^{D \times |\mathbb{P}|}$ and $M_O \in \mathbb{R}^{D \times |\mathbb{P}|}$ respectively. For any numeral, we can compute its input and output embeddings by taking a weighted average of the prototype input and output embeddings respectively based on Eq.1 and use them in exactly the same way as the embeddings of non-numerical words to compute the learning objective (Eq.6). When drawing negative samples, we first set the ratio of numerals and non-numerical words to their actual ratio in the training corpus, to guarantee a sufficient number of numeral negative samples. Then we sample numerals and non-numerical words separately from their respective distributions in the training corpus raised to the power of $\frac{3}{4}$. During training, we optimize the objective function Eq.6 by back-propagating the gradient of error to update both the embedding matrices both the non-numerical word embedding matrices $E_I$, $E_O$ and the prototype number embedding matrices $M_I$, $M_O$. In this way, the embeddings of non-numerical words and numerals are learned jointly in the same space. We show an example in Figure 1.

|  | Non-numerical Word Vocabulary | Numeral Vocabulary |
|---|---|---|
| SOM, GMM, D-LSTM, Fixed | {In-vocab word}, UNK$_{word}$ | all numerals |
| NumAsTok | {In-vocab word}, UNK$_{word}$ | {In-vocab numerals}, UNK$_{num}$ |

Table 1: Vocabularies of different methods.

| Methods | WS353 | MEN | SIM999 |
|---|---|---|---|
| SOM | 64.40 | 71.79 | 36.09 |
| GMM | 64.90 | 71.89 | 36.29 |
| NumAsTok | 65.30 | 71.83 | 35.85 |
| D-LSTM | 63.60 | 71.82 | 34.58 |
| Fixed | 64.35 | 72.17 | 36.27 |
| SG GoogleNews-100B | 70.00 | 74.10 | 44.20 |
| GloVe Wiki-6B | 52.20 | 73.70 | 37.10 |

Table 2: Results on word similarity tasks trained on Wiki-1B. For reference, we also show the results of the official skip-gram and GloVe trained on larger corpora.

## 4 EXPERIMENTS AND RESULTS

We evaluate our methods on four intrinsic and extrinsic tasks: word similarity, embedding numeracy, numeral prediction, and sequence labeling. We report results of our methods based on SOM and GMM separately. We choose the hyper-parameters (e.g., the number of prototypes, GMM initialization and training methods) using validation sets and report the best hyper-parameters for each experiment in Appendix B.

### 4.1 BASELINES

**NumAsTok**  This baseline treats numerals and non-numerical words in the same way, which is very similar to the original skip-gram. The vocabulary includes both high-frequency words and high-frequency numerals. OOV non-numerical words are replaced with symbol UNK$_{word}$ and OOV numerals are replaced with symbol UNK$_{num}$.

**D-LSTM**  Character-level RNNs are often used to encode OOV words (Graves, 2013). Here we apply an LSTM (Hochreiter & Schmidhuber, 1997) to the digit sequence of a numeral and use the last hidden state of the LSTM as the embedding of the numeral. We use the embedding to compute the skip-gram objective function and propagate the gradients back to update the LSTM. The vocabulary of digits is: {0-9, '.', '+', '−', 'e'}.

**Fixed**  This baseline fixed embeddings for numerals with no training. We define the embedding a numeral with value $n$ as $[f(n); \mathbf{1}]/Z$ where $f$ is the squashing function defined in Eq.2, $\mathbf{1} \in \mathbb{R}^{D-1}$ is an all-ones vector, and $Z$ is a constant used to keep the vector norm close to those of non-numerical words and is set to $2 \times D$ by default.

We compare the vocabularies of different methods in Table 1. Our methods, D-LSTM, and Fixed have finite non-numerical vocabularies but infinite numeral vocabularies. In contrast, the NumAsTok baseline has a finite numeral vocabulary and treats all the OOV numerals as UNK$_{num}$.

### 4.2 WORD SIMILARITY FOR NON-NUMERICAL WORDS

To ensure that our methods can still generate high quality embeddings for non-numerical words, we evaluate our trained embeddings on classical intrinsic word similarity tasks, including WordSim-353, (Finkelstein et al., 2001), MEN (Bruni et al., 2014) and Simplex-999 (Hill et al., 2014). We train 300-dimensional word embeddings on the 1B Wikipedia dump and set the context window size to 5, the number of negative samples to 5, and the vocabulary size to $3 \times 10^5$. We use the evaluation tools[1] provided by Jastrzebski (Jastrzebski et al., 2017). Note that while the training data contains numerals, the evaluation tasks do not involve numerals and are only designed to evaluate quality of non-numerical word embeddings. The results are shown in Table 2.

It can be seen that our methods can achieve scores comparable to those of the baselines. The performance of SG trained on 100B GoogleNews is much better than all the other methods probably because of its much larger training corpus. The results show that adding our numeral embedding methods into skip-gram does not harm the quality of non-numerical word embeddings. Additional results of our methods can be found in Appendix C.

---

[1]https://github.com/kudkudak/word-embeddings-benchmarks

| Metrics | Magnitude | | | | Numeration | | | |
|---|---|---|---|---|---|---|---|---|
| | OVA | SC | BC | AVGR | OVA | SC | BC | AVGR |
| SOM | **67.72** | **71.86** | 99.40 | 15.91 | 3.54 | **62.83** | **100.00** | 28.98 |
| GMM | 57.86 | 58.63 | **100.00** | 1.75 | **4.42** | **65.49** | **100.00** | **25.97** |
| NumAsTok | 12.17 | 51.02 | 95.99 | 144.13 | **7.08** | 61.95 | 99.12 | **27.08** |
| D-LSTM | 7.26 | 51.79 | 92.83 | 158.82 | 1.77 | 54.87 | 89.38 | 53.55 |
| Fixed | **83.90** | **78.22** | **100.00** | **1.17** | 0.89 | 49.56 | 99.12 | 56.00 |

Table 3: Magnitude and numeration evaluation results for our methods and baselines. Accuracies of OVA, SC and BC are expressed as percentages. Lower AVGR indicates better performance. Numbers indicating top-2 performance are highlighted.

## 4.3 MAGNITUDE AND NUMERATION OF EMBEDDINGS

Naik et al. (2019) propose a framework for evaluating the ability of numeral embeddings to capture magnitude and numeration. Given a target numeral, its embedding is evaluated against a set of numerals using the **OVA (One-vs-All)**, **SC (Strict Contrastive)** and **BC (Broad Contrastive)** tests:

- **OVA**: The embedding vector distance between the target and its nearest neighbor on the number line should be smaller than that between the target and any other numeral in the set.

- **SC**: The embedding vector distance between the target and its nearest neighbor on the number line should be smaller than that between the target and its second nearest neighbors on the number line.

- **BC**: The embedding vector distance between the target and its nearest neighbor on the number line should be smaller than that between the target and its furthest neighbors on the number line.

We follow the settings described by Naik et al. (2019): for the magnitude evaluation, we run the tests using a set of 2342 numerals that are most frequent in Wikipedia-1B, whose embeddings are well learned by all the methods; and for the numeration evaluation, we run the tests using 113 English words that represent numbers (e.g., 'three', 'billion') sampled from the same corpus and we measure the distance between the target numeral embedding and the word embeddings of these words. We report the accuracy of various embedding models on these three tests, along with the average rank (denoted as **AVGR**) of the target numeral's nearest neighbor among all the candidates based on their vector distances to the target. We use the embeddings trained on Wikipedia-1B.

Table 3 shows the results. The Fixed baseline has the best performance in the magnitude evaluation, which is unsurprising because the numeral embedding vector explicitly contains the (squashed) magnitude. NumAsTok performs very well in the numeration evaluation, which is because the number-representing words used in the evaluation are high-frequency words and their embeddings are adequately trained. Except for these two special cases, our methods can be seen to outperform the baselines with a large margin.

Wallace et al. (2019) recently show that classic embeddings of numerals may contain magnitude information that can be extracted by neural networks. Following their methodology, we conduct two probing tests on our 2342 numerals using multi-layer perceptrons and bilinear functions and then use the resulting models to predict distances between numerals in the **OVA**, **SC**, and **BC** tasks. The results again show the advantage of our methods over the baselines. See Appendix D for details.

## 4.4 NUMERAL PREDICTION

To evaluate the quality of numeral embeddings, we design a new numeral prediction task: choosing the right numeral from a set of candidates given the context of the numeral in a sentence.

We randomly sample 2000 sentences containing numerals from a subset of Wikipedia that is not used in training, with 600 for validation and 1400 for testing. For each sentence, we use the five words preceding and following the target numeral as its context. An example is shown below, where the ten bold words are the context and 2.31 is the target numeral.

In Hollywood, **the average household size was [2.31] and the average family size** was 3.00.

| Metrics | Wikipedia-1B, dim 300 | | | | | | Numeracy-600k, dim 300 | | | | | |
|---|---|---|---|---|---|---|---|---|---|---|---|---|
| | $\mathbf{S_A}$ | | | $\mathbf{S_B}$ | | | $\mathbf{S_A}$ | | | $\mathbf{S_B}$ | | |
| | AVGR | MdAE | MdAPE | AVGR | MdAE | MdAPE | AVGR | Micro-F1 | Macro-F1 | AVGR | Micro-F1 | Macro-F1 |
| SOM | 381.41 | **825.79** | 0.9836 | 455.01 | 1184.60 | 0.9880 | 2.91 | 37.99 | 13.50 | **2.02** | 42.74 | 13.66 |
| GMM | **343.50** | 1184.85 | 0.9450 | **444.15** | 1081.50 | **0.9866** | 2.19 | 41.86 | 18.47 | 2.02 | **44.07** | **13.77** |
| NumAsTok | 600.17 | 1918.00 | 0.9965 | 600.28 | 32772.50 | 19.07 | 4.21 | 9.74 | 5.47 | 6.16 | 24.28 | 4.88 |
| D-LSTM | 357.45 | 1310.65 | **0.9369** | 466.81 | **1080.5** | 0.9908 | 3.98 | 27.98 | 8.80 | 4.49 | 16.49 | 8.42 |
| Fixed | 685.58 | 50371.50 | 42.82 | 672.47 | 50525.00 | 61.59 | 3.23 | 0.00 | 0.01 | 3.23 | 0.00 | 0.00 |

Table 4: The results of the numeral prediction tasks.

We use all the 1400 numerals in the test set as the candidates from which one has to select the right numeral for each test sentence. Given the learned word and numeral embeddings, we define two score functions to rank candidate numerals given the context. Following the skip-gram model, we first define the score of center numeral $n$ predicting context word $c_j$ as $s(c_j|n) = v_{c_j}^{o}{}^{T} v_n^{i}$ and the score of context word $c_j$ predicting the center numeral $n$ as $s(n|c_j) = v_n^{o}{}^{T} v_{c_j}^{i}$. Our first candidate-ranking score function $\mathbf{S_A}$ is the sum of log probabilities of center numeral $n$ predicting each context word $c_j$. We use softmax here to calculate the probability.

$$\mathbf{S_A}(n) = \sum_j \log p(c_j|n) \approx \sum_j \log \frac{e^{s(c_j|n)}}{\sum_{c_k \in \mathbb{V}_t} e^{s(c_k|n)}} = \sum_j s(c_j|n) - \sum_j \log Z(n) \tag{7}$$

where $\mathbb{V}_t$ is the vocabulary of non-numerical words and $Z(n)$ is the normalization factor. The other candidate-ranking score function $\mathbf{S_B}$ is the sum of log probabilities of each context word $c_j$ predicting center numeral $n$.

$$\mathbf{S_B}(n) = \sum_j \log p(n|c_j) \approx \sum_j \log \frac{e^{s(n|c_j)}}{\sum_{n_k \in \mathbb{V}_n} e^{s(n_k|c_j)}} = \sum_j s(n|c_j) - \text{Constant} \tag{8}$$

where $\mathbb{V}_n$ is the set of numerals in the dataset. There are a few other possible score functions, but we find that they lead to results similar to $\mathbf{S_A}$ and $\mathbf{S_B}$.

We use three metrics to evaluate numeral prediction (Spithourakis & Riedel, 2018). **MdAE** is the median of the absolute errors between the predicted and true numerals, **MdAPE** is the median of the absolute percentage errors between the predicted and true numerals, and **AVGR** is the average rank of the true numeral among the candidates. Detailed formulas of the three metrics are shown in Appendix E.

We train embeddings on Wikipedia-1B and report the evaluation results in the left part of Table 4. Our methods significantly outperform the NumAsTok and Fixed baselines on all the three metrics. D-LSTM also performs well but needs more parameters and computing time than our methods.

We also conduct a slightly different numeral prediction task on the recently released Numeracy-600K dataset (the Article Title part) (Chen et al., 2019). This dataset contains 600k sentences with numerals and in each sentence, one numeral is selected and tagged with its order of magnitude. There are eight possible orders of magnitude and the goal is to predict the correct one for the target numeral from its context. To solve this multi-class classification problem, we sample 100 numerals for each order of magnitude and use the mean of their numeral embeddings to create a 'meta' embedding; we then use these 'meta' embeddings to replace the numeral embeddings in the score functions $\mathbf{S_A}$ and $\mathbf{S_B}$ and the highest-scoring order of magnitude is returned.

We split the dataset to 450k sentences for training, 50k for validation and 100k for testing. We use micro-F1 and macro-F1 in addition to **AVGR** as the evaluation metrics. The result is shown in the right part of Table 4. The result shows that our methods achieve much better performance compared to the baselines.

### 4.5 SEQUENCE LABELING ON CUSTOMER SERVICE DATA

To verify the effectiveness of our methods in practice, we evaluate our methods with a sequence labeling task on a dataset of customer service chat log from an online apparel shopping website.

| | | Original | | | | Augmented | | | | Hard | | | |
| --- | --- | --- | --- | --- | --- | --- | --- | --- | --- | --- | --- | --- | --- |
| | | Acc | P | R | F1 | Acc | P | R | F1 | Acc | P | R | F1 |
| | GMM | **97.12** | 91.19 | **90.46** | **90.83** | 97.02 | **91.28** | 90.18 | 90.72 | **96.19** | 86.66 | 85.91 | **86.28** |
| | SOM | 97.04 | 90.74 | 90.45 | 90.60 | **97.03** | 91.19 | **90.43** | **90.81** | 96.06 | 86.18 | **85.93** | 86.06 |
| 100% | D-LSTM | 96.72 | 89.84 | 88.80 | 89.32 | 96.72 | 90.40 | 88.99 | 89.69 | 95.52 | 84.19 | 83.30 | 83.74 |
| | Fixed | 95.75 | 86.19 | 87.42 | 86.80 | 95.86 | 87.13 | 87.65 | 87.39 | 93.97 | 78.39 | 80.18 | 79.27 |
| | NumAsTok | 96.88 | **91.37** | 89.29 | 90.32 | 96.36 | 90.99 | 87.39 | 89.15 | 96.00 | **87.11** | 85.12 | 86.10 |
| | GMM | **96.21** | **89.55** | 86.07 | 87.78 | **95.92** | 89.07 | **85.33** | **87.16** | **95.27** | 84.42 | **81.62** | **82.99** |
| | SOM | 96.20 | 89.50 | **86.18** | **87.81** | 95.88 | **89.12** | 85.29 | **87.16** | 95.23 | **84.44** | 81.50 | 82.94 |
| 30% | D-LSTM | 95.55 | 86.83 | 83.88 | 85.33 | 95.30 | 86.22 | 83.13 | 84.64 | 94.32 | 80.10 | 78.17 | 79.12 |
| | Fixed | 94.67 | 83.51 | 82.69 | 83.10 | 94.48 | 83.40 | 82.02 | 82.71 | 92.92 | 75.03 | 75.18 | 75.10 |
| | NumAsTok | 95.58 | 89.18 | 83.55 | 86.27 | 94.57 | 88.39 | 79.94 | 83.95 | 94.65 | 84.42 | 79.06 | 81.65 |
| | GMM | 93.43 | **82.36** | 75.01 | **78.51** | 92.78 | **81.48** | 72.85 | **76.92** | 93.19 | **80.26** | 72.71 | **76.30** |
| | SOM | **93.48** | 82.13 | **75.11** | 78.46 | **92.87** | 80.96 | **73.22** | 76.89 | **93.24** | 79.47 | **73.04** | 76.11 |
| 10% | D-LSTM | 92.53 | 77.71 | 71.45 | 74.45 | 91.99 | 76.24 | 69.96 | 72.96 | 92.10 | 73.26 | 68.72 | 70.92 |
| | Fixed | 91.90 | 75.39 | 71.41 | 73.34 | 91.48 | 73.96 | 70.20 | 72.02 | 91.06 | 69.50 | 67.47 | 68.46 |
| | NumAsTok | 92.31 | 81.98 | 70.51 | 75.81 | 90.77 | 80.10 | 64.95 | 71.73 | 92.00 | 79.64 | 67.95 | 73.32 |

Table 5: The results of sequence labeling. We report the accuracy, precision, recall, F1 score for the original, augmented, and harder test sets with different training data sizes. Accuracy is in the token level and the other metrics are in the entity level. We report the standard deviations in Appendix H.

This dataset contains a large number of numerals related to height, weight, foot length, etc., and therefore is a good testbed for evaluating numeral embeddings.

The task is to assign a label to each word or numeral in the dataset indicating its information type. We shows two examples below:

W O H O O O O O     W H O O O
82 kg 177 cm what size shall I choose     82 177 what size ?

W, H, O are labels representing weight, height and ordinary word respectively. We show the statistics of the dataset in Appendix G. In order to better evaluate the generalizability, we create two additional test sets. The first one is created by 'augmenting' the original test set with new sentences containing slightly perturbed numerals. For example, we can create new sentences by replacing '177' in the above example with '176' and '178'. The second one contains 'hard' sentences from the original test set that do not have explicit cues for label prediction. For example, the first sentence above contains 'kg' and 'cm' that can greatly facilitate the prediction of W and H, but the second sentence above does not contain such cues and hence is a 'hard' sentence. More details about the two test sets can be found in Appendix F. Finally, we also test the low-resource settings in which only 30% or 10% of the training set is used.

We learn embeddings from the training set using our methods and the baselines and use a validation set to do model selection. We plug the learned embeddings into the Neural-CRF model (Yang & Zhang, 2018) [2] to do sequence labeling without using part-of-speech and character-level features and embedding fine-tuning.

The results are shown in Table 5. Our methods consistently outperform all the baselines on the Accuracy, Recall, and F1 metrics in different configurations. NumAsTok trained with 100% training samples has the highest precision on the original and hard test sets probably because it learns high-quality embeddings for high-frequency numerals included in its vocabulary; but its recall is lower than that of our methods, most likely because of its numeral OOV problem. Comparing the results on the original and augmented test sets, we see that NumAsTok shows a more significant drop in performance than the other methods, which suggests that NumAsTok does not generalize well because of the numeral OOV problem. In the low-resource settings, the advantage of our methods over the baselines becomes even larger, indicating better generalizability and less annotation required for our methods to achieve a promising performance.

## 5 CONCLUSION

In this paper, we propose two novel numeral embedding methods that represent the embedding of a numeral as a weighted average of a set of prototype numeral embeddings. The methods can be

---

[2] https://github.com/jiesutd/NCRFpp

integrated into traditional word embedding approaches such as skip-gram for training. We evaluate our methods on four intrinsic and extrinsic tasks, including word similarity, embedding numeracy, numeral prediction, and sequence labeling, and show that our methods can improve the performance of numeral-related tasks and has better generalizability. Our code and sample data can be found at `path/to/code/`.

An important future direction is to handle numeral polysemy. For example, the numeral "2019" may denote either a year or an ordinary number. One potential method is to assign a different embedding to each sense of a numeral. In this way, "2019" would have one embedding for representing a year and another for representing an ordinary quantity. The similarity function would treat different senses of a numeral differently. For example, the year sense of "2019" would be similar to the year sense of "19" but dissimilar to the sole sense of "2019.5", while the quantity sense of "2019" would be similar to that of "2019.5".

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

## A    GMM INITIALIZATION

Both EM and hard-EM are sensitive to initialization and we use the initialization methods described in (Blömer & Bujna, 2013). We first initialize the mean $\mu_k$ of the $k$-th Gaussian component using one of the following three strategies:

**Random initialization**: choose $\mu_k$ from $X$ randomly. This is suitable when $X$ contains a wide range of numbers, e.g., numbers collected from Wikipedia.

**SOM-based initialization**: initialize $\mu_k$ to $p_k \in \mathbb{P}$ produced by the SOM method.

**K-means initialization**: run randomly initialized k-means on $X$ and then use k-means centroids to initialize $\mu_k$.

We then assign the data samples to their closest means. The standard deviation of the data samples assigned to the $k$-th mean becomes $\sigma_k$.

## B    HYPER-PARAMETERS

We list all of the important hyper-parameters we tune for each model.

**General hyper-parameters**: embedding dimension, context window size, SGD learning rate, batch size, vocabulary size, etc.

**SOM hyper-parameters**: number of prototypes, stage of applying the log-squashing function (stage 1: before prototype induction; stage 2: only in the similarity function).

**GMM hyper-parameters**: number of prototypes, whether we apply the log-squashing function to the numerals, EM initialization (from SOM, random initialization, or k-means initialization), type of EM (hard-EM or soft-EM).

We show the values of the SOM and GMM hyper-parameters in Table 6 and the values of the general hyper-parameters of all the methods in Table 7. We find that the general hyper-parameters influence the performance of our methods and the baselines in the same way, so in most cases, these hyper-parameters are set to be identical for all the methods. For large training corpora (Wiki1B, Numeracy-600k), we use 2048 as the batch size for D-LSTM, because D-LSTM consumes much more GPU memory. We set the batch size of the other methods to 4096. For the sequence labeling tasks, because the data is relatively small and confined to a very specific domain (chat log from online apparel shops), we set a small vocabulary size of 500 for all the methods except NumAsTok and set the vocabulary size of NumAsTok to 550 to ensure that different methods have similar numbers of parameters for word embedding training. Consequently, our methods have $(500 + |\mathbb{P}|) \times D$ parameters for word embedding training and NumAsTok has $550 \times D$ parameters, where $\mathbb{P}$ is the prototype set, whose size is typically smaller than 50, and $D$ is the embedding dimension.

Table 6 also shows that the optimal number of prototypes is around 200–500 for the Wiki1B corpus and 10–25 for the much smaller sequence labeling dataset. As a rule of thumb, we suggest setting the number of prototypes to $(\log N)^2$, where $N$ is the number of distinct numerals in the training corpus.

| | SOM | | GMM | | | |
|---|---|---|---|---|---|---|
| | prototype number | log transform stage | prototype number | log transform | initialization | EM |
| Word similarity (Wiki1B) | 200 | dataset | 200 | True | random | hard |
| Magnitude (Wiki1B) | 200 | dataset | 300 | True | random | soft |
| Numeration (Wiki1B) | 300 | similarity function | 500 | True | random | soft |
| Numeral Prediction (Wiki1B) | 300 | similarity function | 300 | False | random | hard |
| Numeral Prediction (Numeracy-600k) | 50 | dataset | 200 | False | random | hard |
| Sequence Labeling 100 % | 15 | dataset | 30 | False | random | soft |
| Sequence Labeling 30 % | 10 | dataset | 15 | False | k-means | soft |
| Sequence Labeling 10 % | 25 | similarity function | 20 | False | from-som | soft |

Table 6: Hyper-parameter values for GMM and SOM based methods for each experiment.

| | embedding dim | context window | negative samples | epoch | batch size | learning rate | vocabulary size |
|---|---|---|---|---|---|---|---|
| Word similarity (Wiki1B) | 300 | 5 | 5 | 1 | 4096, 2048 | $5 \times 10^{-3}$ | $3 \times 10^5$ |
| Magnitude (**-MAG**) (Wiki1B) | 300 | 5 | 5 | 1 | 4096, 2048 | $5 \times 10^{-3}$ | $3 \times 10^5$ |
| Numeration (**-NUM**) (Wiki1B) | 300 | 5 | 5 | 1 | 4096, 2048 | $5 \times 10^{-3}$ | $3 \times 10^5$ |
| Numeral Prediction (Wiki1B) | 300 | 5 | 5 | 1 | 4096, 2048 | $5 \times 10^{-3}$ | $3 \times 10^5$ |
| Numeral Prediction (Numeracy-600k) | 300 | 2 | 5 | 10 | 4096, 2048 | $5 \times 10^{-3}$ | $1 \times 10^5$ |
| Sequence Labeling 100% 30% 10% | 50 | 2 | 5 | 10 | 50 | $5 \times 10^{-2}$ | 500, 550 |

Table 7: Values of general hyper-parameters for each experiment.

## C  MORE RESULTS ON WIKIPEDIA-1B

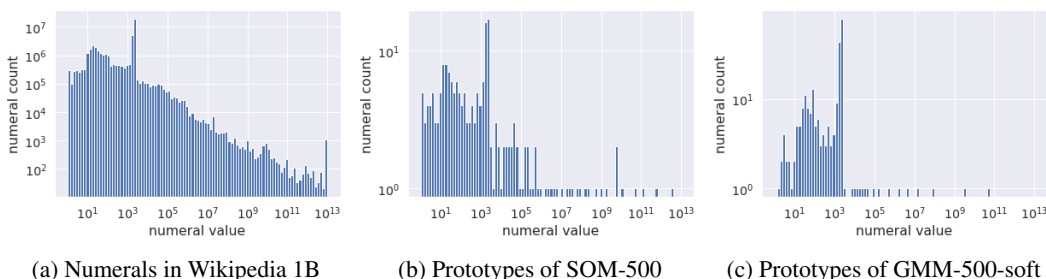

(a) Numerals in Wikipedia 1B  (b) Prototypes of SOM-500  (c) Prototypes of GMM-500-soft

Figure 2: Histograms of numerals and learned prototypes that range from 0 to $10^{13}$. The horizontal axis represents the numeral quantity and the vertical axis represents the number of occurrences, '500' means the number of prototypes, 'soft' means soft-EM.

We show the histograms of numerals in the Wikipedia-1B dataset and the prototypes learned by SOM and GMM in Fig.2. It can be seen that the prototypes induced by our methods have a similar distribution compared to the original numerals.

We also show some examples of prototypes and their nearest non-numerical words in Table 8. We use the embedding trained by the SOM model with 200 prototypes on Wikipedia-1B, and use log transformation in the similarity function.

In addition, we select several typical numerals and non-numerical words and project their embeddings to 2D using t-SNE (Maaten & Hinton, 2008) (Figure 3). We use embeddings learned on Wikipedia-1B corpus using the SOM and GMM methods. The examples and the figures show that our model does capture some semantic relations between numeral quantities and normal words.

We show the training speed of each embedding method on the Wikipedia-1B dataset in Table 9. The batch size is set to 2048 for all the methods. Our methods are slower than NumAsTok but are faster than D-LSTM.

| Prototype Value | Most Related Non-numerical Words |
|---|---|
| 8186446.58 | million, billion, total, budget, funding, estimated, dollars |
| 10372.49 | thousand, approximately, thousands, millions, roughly, hundreds |
| 2000.06 | nearly, millennium, decade, internet, twentieth, worldwide, latest |
| 1598.79 | johann, renaissance, giovanni, dutch, baroque, vii, shakespeare |
| 10.00 | ten, six, eleven, pm, seconds, eight |

Table 8: Examples of prototypes and their nearest non-numerical words.

| Method | SOM | GMM | NumAsTok | D-LSTM | Fixed |
|---|---|---|---|---|---|
| Speed (sent/s) | 13590.93 | 12691.18 | **22907.97** | 8421.66 | 13055.08 |

Table 9: Training speed for each methods.

# D  PROBING TESTS

We apply two probing tests using neural network on our methods and baselines in order to compare their ability to encode magnitude information in a non-linear way. The first test is Decoding (predicting the numeral value from its embedding using MLP). The second is Subtraction (predicting the difference between two numerals from their embeddings using MLP or BiLinear functions). We illustrate the tasks and the models we use in Figure 4.

We first create the datasets for the two probing tests based on the dataset from the magnitude evaluation of Section 4.3 (containing 2342 numerals). For Decoding, the dataset can be directly used. For Subtraction, we randomly sample $10^5$ pairs of numerals $(n_1, n_2)$ from the dataset and assign $n_1 - n_2$ as the prediction target. Following Wallace et al. (2019), we randomly split $80\%$ of each dataset for training and $20\%$ for testing. We use SGD to optimize the mean square error (MSE) loss. We report the root-mean-square error (RMSE) results for the two tasks in Table 10.

| | Decoding | | Subtraction | | | |
|---|---|---|---|---|---|---|
| | MLP1 | MLP2 | BiLinear | BiLinear+MLP1 | MLP1 | MLP2 |
| GMM | **77.73** | **68.29** | **1006.66** | **62.04** | 18.10 | **4.40** |
| SOM | 289.40 | 182.91 | 2109.16 | 100.03 | 33.62 | 14.04 |
| NumAsTok | 1027.33 | 1035.25 | 2835.32 | 282.32 | **15.52** | 4.66 |
| D-LSTM | 1433.45 | 1397.00 | 3208.96 | 634.20 | 717.04 | 644.41 |
| Fixed | 2220.75 | 2225.31 | 3791.96 | 3790.93 | 3791.91 | 3791.98 |

Table 10: Probing test results. MLP1 and MLP2 denote MLP with one and two hidden layers respectively.

The results show that our two methods are significantly better than the baselines on Decoding. On Subtraction, they are better than the baselines when using BiLinear and are comparable to NumAsTok but much better than the other baselines when using MLP. We found that the performance is very sensitive to the neural network architecture and MLP with two hidden layers performs best.

We then use the MLP2 models trained on Subtraction to determine the distance between two numerals when conducting the magnitude evaluation of Section 4.3. The results are shown in Table 11. The results show that our methods have better performance than the baselines overall. One interesting observation is that, although our SOM based method has worse RMSE than NumAsTok as shown in Table 10, it outperforms NumAsTok in the magnitude evaluation.

# E  NUMERAL PREDICTION EVALUATION METRICS

We denote the target numeral by $n_i$, the numeral with the highest ranking score by $\hat{n}_i$, and the rank of the target numeral by $r_i$. The error $e_i$ and percentage error $pe_i$ can be calculated as:

$$e_i = n_i - \hat{n}, \qquad pe_i = \frac{n_i - \hat{n}_i}{n_i} \qquad (9)$$

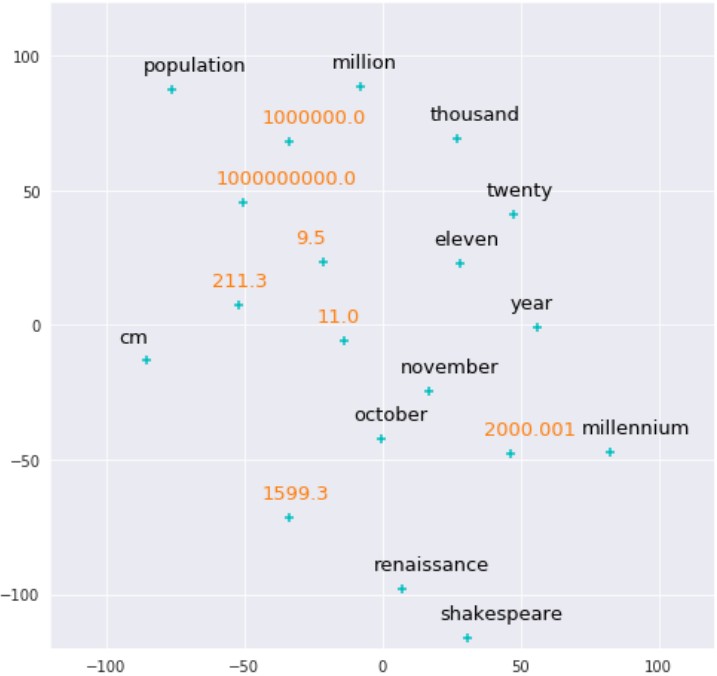

(a) t-SNE plot for embedding trained by the SOM-based method with 200 prototypes.

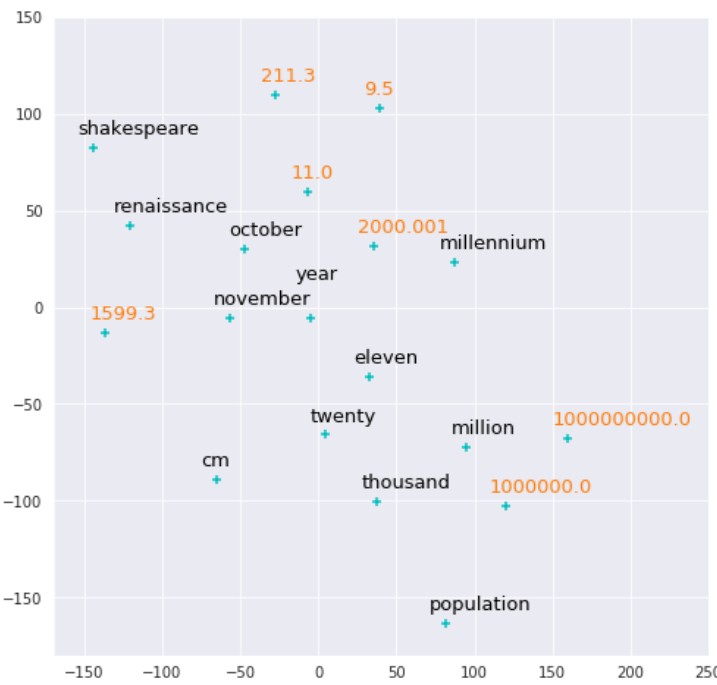

(b) t-SNE plot for embedding trained by the GMM-based method with 300 prototypes, random initialization and soft-EM training.

Figure 3: 2D t-SNE results for the SOM-based and GMM-based methods.

Then we use the median of the absolute errors, the median of the absolute percentage errors, and the average rank as the evaluation metrics.

$$MdAE = \text{median}\{|e_i|\}, \qquad MdAPE = \text{median}\{|pe_i|\}, \qquad AVGR = \overline{r_i} \qquad (10)$$

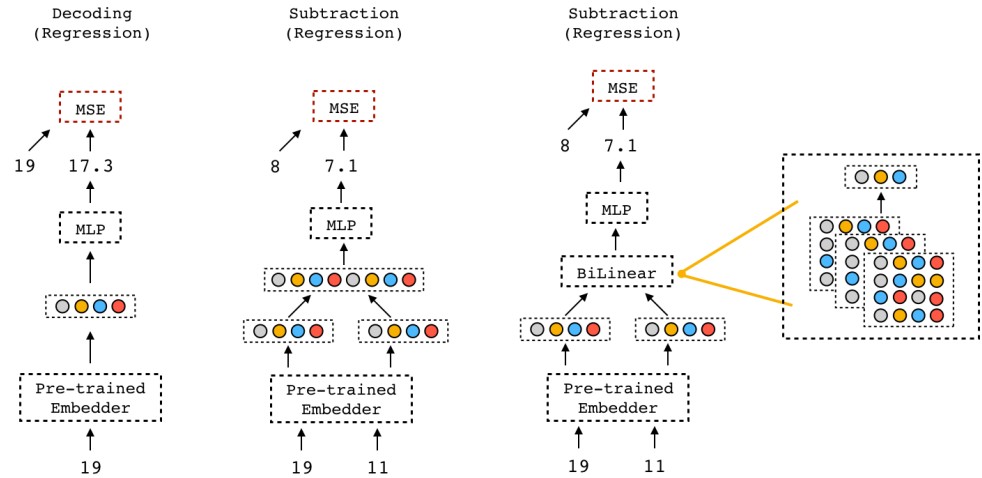

Figure 4: Diagrams of the probing models for Decoding and Subtraction.

|          | OVA  | SC    | BC     | AVGR    |
|----------|------|-------|--------|---------|
| GMM      | **8.03** | 48.76 | **100.00** | **16.45**   |
| SOM      | 4.40 | **57.43** | **100.00** | 59.41   |
| NumAsTok | 1.96 | 50.90 | **100.00** | 106.98  |
| D-LSTM   | 0.51 | 52.09 | **100.00** | 545.87  |
| Fixed    | 0.00 | 00.00 | 0.00   | 1170.68 |

Table 11: Results of the magnitude evaluation using MLP2 trained on the Subtraction probing task. Accuracies of OVA, SC and BC are expressed as percentages. Lower AVGR indicates better performance.

## F    AUGMENTED AND HARD TEST SETS IN SEQUENCE LABELING

The augmented test set is created by reasonably perturbing the numerals in a sentence. For example, for a numeral '173' that describes height, we generate new samples by changing '173' to '174' or '175' while keeping the other non-numerical words in the sentence unchanged. For a decimal such as '1.7 meters', we change it to '1.6' or '1.8'. The perturbation will not change the decimal places of numerals and will only change the quantity slightly, which makes the generated sentences reasonable.

The hard test set is created by manually collect 'hard' samples in the original test set. Hard samples do not have explicit patterns, meaning that a numeral's tag cannot be easily inferred by its adjacent words. For example, tags of numerals followed by units like 'cm', 'm', 'kg', 'years' and 'feet' can be figured out easily, so we exclude them from the hard test set. Customers are very likely to use ambiguous expressions like: 'I'm 16.5, can I buy 24?', where 16.5 is about foot length and 24 is the shoe size. These ambiguous sentences are included in the hard test set.

## G    STATISTICS OF SEQUENCE LABELING DATASET

We show the statistics of the customer-service dataset in the Table 12. The vocabulary is small because the dataset is confined to a specific domain: online customer service chat log about apparel purchase. In this dataset, most of the sentences are about sizes of various kinds of clothes and are very short and ambiguous.

## H    SEQUENCE LABELING RESULT WITH STANDARD DEVIATION.

| | Number of Sentences | | | |
|---|---|---|---|---|
| Train | Dev | Original Test | Augmented Test | Hard Test |
| 1389 | 793 | 1802 | 8052 | 726 |
| | Statistics of Training Set | | | |
| Token Vocab | Numeral Vocab | Avg sent length | Numeral Ratio | labels |
| 505 | 234 | 10.42 | 15.89 % | 21 |

Table 12: Statistics of low-resource customer-service dataset.

| | | Original | | | | Augmented | | | | Hard | | | |
|---|---|---|---|---|---|---|---|---|---|---|---|---|---|
| | | Acc | P | R | F1 | Acc | P | R | F1 | Acc | P | R | F1 |
| 100% | GMM | **97.12** ±0.05 | 91.19 ±0.11 | 90.46 ±0.17 | **90.83** ±0.14 | 97.02 ±0.06 | **91.28** ±0.17 | 90.18 ±0.19 | 90.72 ±0.18 | **96.19** ±0.05 | **86.66** ±0.13 | 85.91 ±0.36 | **86.28** ±0.24 |
| | SOM | 97.04 ±0.04 | 90.74 ±0.15 | 90.45 ±0.10 | 90.60 ±0.12 | **97.03** ±0.03 | 91.19 ±0.14 | **90.43** ±0.11 | **90.81** ±0.13 | 96.06 ±0.09 | 86.18 ±0.14 | **85.93** ±0.32 | 86.06 ±0.23 |
| | D-LSTM | 96.72 ±0.06 | 89.84 ±0.26 | 88.80 ±0.24 | 89.32 ±0.25 | 96.72 ±0.07 | 90.40 ±0.29 | 88.99 ±0.23 | 89.69 ±0.26 | 95.52 ±0.14 | 84.19 ±0.66 | 83.30 ±0.55 | 83.74 ±0.60 |
| | Fixed | 95.75 ±0.11 | 86.19 ±0.38 | 87.42 ±0.21 | 86.80 ±0.29 | 95.86 ±0.10 | 87.13 ±0.25 | 87.65 ±0.28 | 87.39 ±0.26 | 93.97 ±0.16 | 78.39 ±0.46 | 80.18 ±0.43 | 79.27 ±0.45 |
| | NumAsTok | 96.88 ±0.07 | **91.37** ±0.40 | 89.29 ±0.09 | 90.32 ±0.21 | 96.36 ±0.05 | 90.99 ±0.41 | 87.39 ±0.09 | 89.15 ±0.20 | 96.00 ±0.10 | **87.11** ±0.52 | 85.12 ±0.03 | 86.10 ±0.26 |
| 30% | GMM | **96.21** ±0.07 | **89.55** ±0.15 | 86.07 ±0.32 | 87.78 ±0.24 | **95.92** ±0.09 | 89.07 ±0.32 | **85.33** ±0.40 | **87.16** ±0.36 | **95.27** ±0.13 | 84.42 ±0.41 | **81.62** ±0.47 | **82.99** ±0.43 |
| | SOM | 96.20 ±0.03 | 89.50 ±0.17 | **86.18** ±0.29 | **87.81** ±0.08 | 95.88 ±0.08 | **89.12** ±0.16 | 85.29 ±0.41 | **87.16** ±0.13 | 95.23 ±0.02 | **84.44** ±0.30 | 81.50 ±0.22 | 82.94 ±0.09 |
| | D-LSTM | 95.55 ±0.08 | 86.83 ±0.29 | 83.88 ±0.36 | 85.33 ±0.32 | 95.30 ±0.12 | 86.22 ±0.41 | 83.13 ±0.50 | 84.64 ±0.46 | 94.32 ±0.07 | 80.10 ±0.33 | 78.17 ±0.39 | 79.12 ±0.35 |
| | Fixed | 94.67 ±0.06 | 83.51 ±0.21 | 82.69 ±0.18 | 83.10 ±0.12 | 94.48 ±0.08 | 83.40 ±0.23 | 82.02 ±0.26 | 82.71 ±0.17 | 92.92 ±0.05 | 75.03 ±0.06 | 75.18 ±0.38 | 75.10 ±0.17 |
| | NumAsTok | 95.58 ±0.03 | 89.18 ±0.25 | 83.55 ±0.31 | 86.27 ±0.10 | 94.57 ±0.07 | 88.39 ±0.40 | 79.94 ±0.16 | 83.95 ±0.21 | 94.65 ±0.03 | 84.42 ±0.39 | 79.06 ±0.23 | 81.65 ±0.10 |
| 10% | GMM | 93.43 ±0.12 | **82.36** ±0.17 | 75.01 ±0.52 | **78.51** ±0.21 | 92.78 ±0.03 | **81.48** ±0.25 | 72.85 ±0.36 | **76.92** ±0.14 | 93.19 ±0.04 | **80.26** ±0.41 | 72.71 ±0.09 | **76.30** ±0.19 |
| | SOM | **93.48** ±0.11 | 82.13 ±0.26 | **75.11** ±0.41 | 78.46 ±0.21 | **92.87** ±0.10 | 80.96 ±0.25 | **73.22** ±0.37 | 76.89 ±0.19 | **93.24** ±0.10 | 79.47 ±0.07 | **73.04** ±0.49 | 76.11 ±0.30 |
| | D-LSTM | 92.53 ±0.19 | 77.71 ±0.38 | 71.45 ±0.82 | 74.45 ±0.61 | 91.99 ±0.26 | 76.24 ±0.40 | 69.96 ±0.11 | 72.96 ±0.80 | 92.10 ±0.16 | 73.26 ±0.26 | 68.72 ±0.70 | 70.92 ±0.40 |
| | Fixed | 91.90 ±0.05 | 75.39 ±0.46 | 71.41 ±0.58 | 73.34 ±0.17 | 91.48 ±0.12 | 73.96 ±0.64 | 70.20 ±0.73 | 72.02 ±0.40 | 91.06 ±0.06 | 69.50 ±0.78 | 67.47 ±0.27 | 68.46 ±0.25 |
| | NumAsTok | 92.31 ±0.12 | 81.98 ±0.44 | 70.51 ±0.56 | 75.81 ±0.29 | 90.77 ±0.14 | 80.10 ±0.65 | 64.95 ±0.66 | 71.73 ±0.23 | 92.00 ±0.06 | 79.64 ±0.64 | 67.95 ±0.38 | 73.32 ±0.20 |

Table 13: The results of sequence labeling. We report the accuracy, precision, recall, F1 score for the original, augmented, and harder test sets with different training data sizes. Accuracy is in the token level and the other metrics are in the entity level.

