# OpenReview forum: "Learning Numeral Embedding"
_ICLR.cc/2020/Conference — Reject_

### Official Review · AnonReviewer2 · 2019-10-16
**Official Blind Review #2**

**Rating:** 6

**Review:**


Summary:
The paper talks about a recently highlighted problem in word embeddings which is their incapability to represent numerals, especially the out-of-vocabulary numerals. For addressing the problem, they propose a method that induces a finite set of prototype numerals using either self-organizing map or Gaussian Mixture model. Then, each numeral is represented as a weighted average of prototype numeral embeddings. The method also involves squashing large quantities using log function. Finally, the training is performed similar to Skip-gram in word2vec but with the embedding of numerals computed using prototype numerals.

Questions:
1. There are two basic motivations of the paper: (1) Most of the word embedding methods do not embed numerals correctly. (2) There is no mechanism of handing OOV numerals. The second one is well addressed but for the former one, it has been shown in recent work [1] that most of the embedding methods do have numerical reasoning capabilities. As stated in [1], the results of [2] demonstrate the opposite conclusion because their analysis is based on cosine distance and nearest neighbor which are not capable of capturing non-linear dependencies between embeddings.

So, it would be great if, for the results in Section 4.3 instead of using cosine distance, some neural models could be utilized for evaluation (3 layer MLP similar to [1]).

2. In Section 4.4, the task is to predict the target numeral given its context words (similar to CBOW) while the embeddings are trained with a modified skip-gram model. Can one expect superior results if one uses modified CBOW for training embeddings rather than skip-gram?

3. In Table 5, with 100% training data, the performance of all the methods is very close. It would be better if mean and variance across multiple runs are reported.

**Experience Assessment:**

I have read many papers in this area.

**Review Assessment: Checking Correctness Of Derivations And Theory:**

I assessed the sensibility of the derivations and theory.

**Review Assessment: Checking Correctness Of Experiments:**

I assessed the sensibility of the experiments.

**Review Assessment: Thoroughness In Paper Reading:**

I read the paper at least twice and used my best judgement in assessing the paper.

---

> ### Author Response · Authors · 2019-11-14
> **Response to Reviewer #2**
>
> We would like to thank the reviewer for detailed review and helpful feedback.
>
> 1.       We suppose [1] is "Do NLP Models Know Numbers? Probing Numeracy in Embeddings [EMNLP19]" and [2] is "Exploring Numeracy in Word Embeddings [ACL19]".
>
> [1] is a concurrent work that was made public right before we submitted our paper, so we were not able to consider its findings in our experimental setup. Following your suggestion, we are testing neural probing/decoding methods and applying them to the tasks in section 4.3.
>
> The first test is Decoding (predicting the numeral value from its embedding using MLP). The second is Subtraction (predicting the difference between two numerals from their embeddings using MLP or BiLinear functions).
>
> The preliminary result shows that, compared with the baselines, our method is much better at Decoding, and better at Subtraction using BiLinear, but slightly worse than NumAsTok (normal word embeddings) at Subtraction using MLP. We find that the probing test result is sensitive to the network structure. We will provide detailed probing results as well as the evaluation results on the tasks (OVA, SC, BC, AVGR) of Section 4.3 using these trained neural networks before the response deadline.
>
> 2.       It is probably hard to say. While the objective of CBOW fits the numeral prediction task better, skip-gram is known to handle infrequent words (hence most of the numerals) better.
>
> 3.      We will add the std of the experimental results in the appendix. We find that in most cases the std is quite small, so the result is stable.

---

### Official Review · AnonReviewer1 · 2019-10-23
**Official Blind Review #1**

**Rating:** 3

**Review:**

I have read the author response.  Thank you for responding to my concerns.

Original review:
This paper presents a word embedding approach for numbers.  The method is based on finding prototype numbers, and then representing numbers as a weighted average of the prototype embeddings, where the weights are based on numeric proximity.  The approach provides some gains over baselines on number similarity, number prediction, and sequence tagging tasks.  While modeling numbers is an interesting task, several aspects of the paper needed more clarification, and the paper’s focus may be somewhat narrow for the ICLR audience.

I think the paper could use a better motivation for the prototype-based approach.  In particular, the fact that the approach uses only quantity (and not the form of numbers) to represent similarity is contrary to my intuition.  For example, I would expect 1960 and 1960.1 to behave very differently in text, because one is a year and the other isn’t.  But the proposed method, if I understand it correctly, would give them similar embeddings because they are very close numerically.

I was not able to understand the SOM portion of the method, it is not self-contained within this paper.

On the Numeracy-600K data set, Chen et al. (2019) shows much higher F1 results than those shown here.  What explains this difference?

The proposed method seems relatively similar to that of (Spithourakis & Riedel, 2018), in that it exposes numeric quantity to the language/embedding model, and uses GMMs to represent numeral distributions.  More clarity about how this approach compares to that one (and others) would be helpful.  Also, how does the (Spithourakis & Riedel, 2018) approach fare on e.g. the number prediction tasks in the submission?  It is true that the submission's approach produces general-purpose embeddings that can be re-used, unlike (Spithourakis & Riedel, 2018).  But we would like to know whether that generality comes at the cost of performance on tasks, and if so how much of a cost.

Minor:
The Lund and Burgess reference seems incorrect.  I don’t see that those authors published a paper by that title in Brain and Cognition in 1996.


**Experience Assessment:**

I have read many papers in this area.

**Review Assessment: Checking Correctness Of Derivations And Theory:**

I assessed the sensibility of the derivations and theory.

**Review Assessment: Checking Correctness Of Experiments:**

I assessed the sensibility of the experiments.

**Review Assessment: Thoroughness In Paper Reading:**

I read the paper at least twice and used my best judgement in assessing the paper.

---

> ### Author Response · Authors · 2019-11-14
> **Response to Reviewer #1**
>
> We would like to thank the reviewer for the detailed review and helpful feedback.
>
> [uses only quantity to represent similarity] Our current similarity function is solely based on quantity, which is not perfect but serves as a solid start point for future extension. Regarding the 1960 vs. 1960.1 problem, we observe that numerals like 1960.1 are rare in practice and hence our methods would work in most cases. A potential solution for this problem is to treat each numeral as having multiple senses, each with a different embedding. The numeral 1960 would have one embedding for representing a year and another for representing an ordinary quantity.
>
> [SOM portion of the method] Sorry for the unclearness of the SOM part.  SOM can be viewed as a clustering method except that its cluster centers are distributed more evenly on the number line. We will improve our description of SOM in section 3.2.
>
> [Numeracy-600K F1 scores] Chen et al. (2019) aim to test different neural models' ability to predict the numeral magnitude given the context. They rely on advanced neural architectures (capsule net, CNN, LSTM) to discover patterns in the text that indicate magnitude, and do not care about the embedding (they use random embedding for initialization). While we use their dataset, our goal is different in that we aim to evaluate embeddings instead of neural architectures. Hence, we only use the simple Skipgram model without any complex architecture and additional parameters.  This is why our scores are lower than those in Chen et al. (2019).
>
> [Comparison to Spithourakis & Riedel, 2018]
>  As correctly pointed out by the reviewer, Spithourakis & Riedel, 2018 is a language model that handles numerals, while our method learns general-purpose numeral embeddings that can be used in any neural model. For the number prediction task, we are currently trying to apply the system of Spithourakis & Riedel, 2018 to our dataset. However, because of the complexity in rewriting their preprocessing code for our dataset, as well as the time needed for training and tuning on the large Wikipedia-1B data, we do not expect to finish the experiments before the response deadline.
>
> Minor:
> Thanks for pointing that out and sorry for the reference mistake. We have updated it.

---

### Official Review · AnonReviewer3 · 2019-10-23
**Official Blind Review #3**

**Rating:** 6

**Review:**


The paper proposes a novel method for embedding numerals which can be learned by using neural word embedding learning techniques. The paper motivates the work by reviewing the difficulty of embedding components to represent numerals: OOV in most cases. Their main contribution is the introduction of a method composes numeral embedding by a weighted average of prototype embeddings based on the similarities between the numeral and prototypes. There are two proposed prototypes: SOM and GMM and the similarity functions are an absolute difference and the density function respectively. During the training, the numerals have the proposed embeddings while the others have normal word embeddings. The paper slightly modifies the negative sampling to ensure numerals being sampled. A series of 4 empirical studies have been presented. First, the paper confirms that the proposed method does not negatively affect non-numeral embeddings. And then, the quality of the numeral embeddings are evaluated and compared. The experiments show that the proposed method has better performance on numerical property tests, numeral prediction, and a sequence labeling task.

Overall, this paper has a novel contribution. The proposed method is well motivated and quite justified by the experiments abliet lacking comparison with previous published results. However, it has some weaknesses.

For the method part, one of the limitations is that it is not an end-to-end method and requires a regular expression to identify the numeral. Second, the weighted average of the prototypes is reasonable, but the similarity function only relies on the magnitude. I think there are other aspects of numerical tokens that it might not be able to capture (e.g. “2019” is similar to “19” in some context). In terms of training, I think it is not hard to extend the method to full language model training (using softmax). However, adding all numerals to the vocabulary would add significant overhead.

For the experiments, some design decisions are left unjustified. For example, a simple ablation experiment on the squashing function will be helpful. Furthermore, guidelines or empirical results on the effect of the number of prototypes can increase the impact of the paper. Finally, I think an analysis of the performance of numerical types will be helpful for future works (e.g., dates, phone numbers, currencies, etc, or discrete vs continuous).

Minor comments and questions:
1. The log sigmoid in equation 6 is a bit strange, isn’t it log sum exp(). https://arxiv.org/pdf/1402.3722v1.pdf
2. Why do you create a new dataset for the experiment in section 4.3?
3. In section 4.4, you rank only numerals in the test set, but the scores are computed based on all numerals in the vocab. Do you have the performance of ranking all numerals?
4. Just to confirm the “training” in section 4.4 refers to learning the embedding using the skip-gram model, right?


**Experience Assessment:**

I have read many papers in this area.

**Review Assessment: Checking Correctness Of Derivations And Theory:**

I assessed the sensibility of the derivations and theory.

**Review Assessment: Checking Correctness Of Experiments:**

I assessed the sensibility of the experiments.

**Review Assessment: Thoroughness In Paper Reading:**

I read the paper at least twice and used my best judgement in assessing the paper.

---

> ### Author Response · Authors · 2019-11-13
> **Response to Reviewer #3**
>
> We would like to thank the reviewer for the detailed review and helpful feedback.
>
> [ Not an end-to-end method and requires a regular expression ]
> We use the standard regular expression for identifying numerals. It is ready-to-use and very fast and accurate, so we choose not to rely on learning for numeral identification. The prototype induction step can potentially be modified so that it can be trained jointly with the embedding training steps in an end-to-end manner. We leave this for future work.
>
> [“2019” is similar to “19”] Thank you for pointing out this problem. Our current similarity function is solely based on the intrinsic property of numbers that numbers close to each other are likely to convey similar semantic information. It is not perfect but serves as a solid start point for future extension. Regarding the 2019 vs. 19 problem, one potential solution is to treat each numeral as having multiple senses, each with a different embedding, so 2019 and 19 would each have one embedding for representing a year and another for representing an ordinary quantity; then we design the similarity function such that their year-representing embeddings have high similarity.
>
> [ Extend to language model training (using softmax)] It is possible to extend our method to language model training, but due to the infinite vocabulary of numerals, using softmax is intractable and the negative sampling technique would be preferred.
>
> [ A simple ablation experiment on the squashing function] If we do not squash the numbers, the gradient becomes NaN during training because of the occurrence of very large numbers like 10^15 in the corpus. This is exactly our motivation for using the squash function. We will clarify this in the paper.
>
> [ Guidelines on the selection of prototype numbers ] Determining the number of prototypes is very similar to determining the number of clusters in clustering. We show the tuned prototype numbers in our experiments in Table 6 in Appendix B. The optimal number of prototypes is around 200--500 for Wiki1B and is 10--25 for the much smaller sequence labeling dataset. So we suggest a rule of thumb that the prototype number is set to $(\log(n))^2$, where $n$ is the number of distinct numerals in the training corpus.
>
> Minor question:
> 1.      We use negative sampling instead of softmax to deal with the infinite numeral vocabulary, so we have sigmoid in our objective. (See the equation at the top of page 4 in https://arxiv.org/pdf/1402.3722v1.pdf )
> 2.      The reason is that the original work does not provide the dataset. We collect our own dataset following their procedure.
> 3.      We do not have the performance for all the numerals. We could measure the performance if time permits, but we expect that the results would follow a similar pattern.
> 4.      Yes.

---

### Author Response · Authors · 2019-11-15
**Short summary for paper revision**

ICLR 2020 Conference Paper762 AuthorsChengyue Jiang(privately revealed to you)
15 Nov 2019 (modified: 15 Nov 2019)ICLR 2020 Conference Paper762 Official CommentReaders:  Everyone
Comment: We have updated our paper in response to the reviews with the following revisions:
1. We described a new experiment of probing tests using neural networks in Section 4.3 and Appendix D.
2. We added why squashing is necessary in Section 3.1.
3. We clarified the SOM method in Section 3.2.
4. We discussed numeral polysemy (e.g., "2019") in Section 5.
5. We added the results of sequence labeling with standard deviations in Appendix H.
6. We changed the reference for Lund & Burgess (1996) and added the reference for t-SNE (Maaten & Hinton, 2008).

In addition, we are still in the process of applying the language model of Spithourakis & Riedel, 2018 to our numeral prediction task and will include the results in the final version of the paper.

---

### Decision · Program_Chairs · 2019-12-19

**Decision:**

Reject

**Comment:**

This paper proposes better methods to handle numerals within word embeddings.

Overall, my impression is that this paper is solid, but not super-exciting. The scope is a little bit limited (to only numbers), and it is not by any means the first paper to handle understanding numbers within word embeddings. A more thorough theoretical and empirical comparison to other methods, e.g. Spithourakis & Riedel (2018) and Chen et al. (2019), could bring the paper a long way.

I think this paper is somewhat borderline, but am recommending not to accept because I feel that the paper could be greatly improved by making the above-mentioned comparisons more complete, and thus this could find a better place as a better paper in a new venue.